# Exploring the Potential of Natural Killer Cell-Based Immunotherapy in Targeting High-Grade Serous Ovarian Carcinomas

**DOI:** 10.3390/vaccines12060677

**Published:** 2024-06-18

**Authors:** Kawaljit Kaur, Jashan Sanghu, Sanaz Memarzadeh, Anahid Jewett

**Affiliations:** 1Division of Oral Biology and Medicine, The Jane and Jerry Weintraub Center for Reconstructive Biotechnology, University of California School of Dentistry, 10833 Le Conte Ave, Los Angeles, CA 90095, USA; ajewett@dentistry.ucla.edu; 2Department of Obstetrics and Gynecology, David Geffen School of Medicine, University of California Los Angeles, Los Angeles, CA 90095, USA; jsanghu@mednet.ucla.edu (J.S.); smemarzadeh@mednet.ucla.edu (S.M.); 3Eli and Edythe Broad Center of Regenerative Medicine and Stem Cell Research, University of California Los Angeles, Los Angeles, CA 90095, USA; 4The Jonsson Comprehensive Cancer Center, University of California Los Angeles, Los Angeles, CA 90095, USA; 5Molecular Biology Institute, University of California Los Angeles, Los Angeles, CA 90095, USA; 6The VA Greater Los Angeles Healthcare System, Los Angeles, CA 90073, USA

**Keywords:** ovarian cancer, high-grade serous ovarian cancers, immunotherapies, cytotoxicity, cytokines, IFN-γ, natural killer cells, super-charged NK cells

## Abstract

High-grade serous ovarian cancers (HGSOCs) likely consist of poorly differentiated stem-like cells (PDSLCs) and differentiated tumor cells. Conventional therapeutics are incapable of completely eradicating PDSLCs, contributing to disease progression and tumor relapse. Primary NK cells are known to effectively lyse PDSLCs, but they exhibit low or minimal cytotoxic potential against well-differentiated tumors. We have introduced and discussed the characteristics of super-charged NK (sNK) cells in this review. sNK cells, in comparison to primary NK cells, exhibit a significantly higher capability for the direct killing of both PDSLCs and well-differentiated tumors. In addition, sNK cells secrete significantly higher levels of cytokines, especially those known to induce the differentiation of tumors. In addition, we propose that a combination of sNK and chemotherapy could be one of the most effective strategies to eliminate the heterogeneous population of ovarian tumors; sNK cells can lyse both PDSLCs and well-differentiated tumors, induce the differentiation of PDSLCs, and could be used in combination with chemotherapy to target both well-differentiated and NK-induced differentiated tumors.

## 1. Introduction and Background: Ovarian Cancer

Ovarian carcinomas are the second most common cause of gynecological malignancies-related death and are composed of a diverse group of tumors based on histological, molecular, and genetic factors [1]. In total, 70% are high-grade serous ovarian cancers (HGSOCs), 10% are endometrioid, 10% is clear cells, 3% mucinous, and approximately 5% low-grade serous carcinoma [1,2,3,4,5]. HGSOCs arise from the epithelium of the distal fallopian tube, are typically diagnosed at an advanced stage, and are responsible for the majority of ovarian cancer-related deaths [2,3,6,7]. The standard approach of treating HGSOCs is surgical intervention combined with platinum-based chemotherapy [8], which initially yields positive responses in many patients [9]. Despite advances in treatment, the prognosis of HGSOC remains poor, and the majority of treated cases relapse within a few years of diagnosis resulting in a 5-year survival rate lower than 40% [7]. Thus, the high recurrence rate and chemoresistance development highlight the urgent need for novel treatment strategies to improve patient prognosis [10]. Recently, the FDA approved mirvetuximab soravtansine as a novel therapeutic alternative for a subset of platinum-resistant patients with high levels of folate receptor alpha expression within the tumor [11,12]. While this therapy provides an alternative treatment for this subpopulation, the observed median post-treatment progression-free survival in cancer patients was 4–5.6 months [11,13], underscoring the urgent need to develop innovative treatment approaches.

Immunotherapy has emerged as a promising approach for cancer treatment, particularly for HGSOC. Natural killer (NK) cells, a type of cytotoxic lymphocyte, play a crucial role in the innate immune system and have demonstrated potential in targeting cancer cells [14,15,16]. This review explores the potential of NK cell-based immunotherapy in targeting HGSOC, focusing on the characteristics and advantages of supercharged NK (sNK) cells.

## 2. Heterogeneity and Mechanism of Chemoresistance in HGSOC

The complexity of treating HGSOCs is exacerbated by the tumor’s heterogeneity [17] and the absence of clearly targetable driver mutations [18]. Studies have demonstrated that most of the mutations in primary tumors persist post chemotherapy suggesting that most clones are chemoresistant [18,19]. It was found that pre-treatment subclones of the tumor population could undergo expansion during chemotherapy or platinum-based therapy resulting in tumor relapse [20,21]. Several studies were performed to investigate the mechanism of chemoresistance in HGSOC. Notably, platinum-based therapies, while effective in eradicating most tumor cells, likely leave behind a population of poorly differentiated stem-like cells implicated in disease progression and tumor relapse [22,23,24,25]. The identification and targeting of these tumor stem-like cells have been explored in various studies and found increased expression levels of cell surface markers like CD24, CD44, CD133, CD117, elevated ALDH activity, and increased levels of pluripotency-related transcription factors such as Nanog, Oct3/4, and Sox2 [26,27,28,29,30,31]. Indeed, the increased expression of pluripotency-related transcription factors is seen in recurrent HGSOC tumor samples compared to chemo-naive ones [32].

It is now well-known that the tumor tissues consist of diverse cellular components including cancer cells and stromal cells [5]. IL-6 was found to induce chemoresistance in ovarian cancer, and mesenchymal stromal cells, particularly cancer-associated fibroblasts (CAFs), were found to be the main source of IL-6 secretion in ovarian cancer [33]. It was found that CCL2 and CCL5 secreted by CAFs could stimulate IL-6 secretion in ovarian cancer cells ultimately resulting in chemoresistance in tumors [34]. Several downstream mechanisms including PYK2, Ras/MEK/ERK, PI3K/Akt, and JAK/STAT3 signaling play a role in IL-6-mediated chemoresistance in ovarian cancers [33,34,35]. Leung et al. demonstrated that a crosstalk signal of CAFs and endothelial cells could lead to chemoresistance in ovarian cancer via regulation of the lipoma preferred partner (LPP) gene in endothelial cells [36]. Stromal cell-derived midkine was found to associated with chemoresistance in ovarian cancer due to increased expression levels of IncRNA ANRIL [37]. Gonzalez et al. performed in-depth single-cell phenotypic characterization of HGSOC using multiparametric mass cytometry (cyTOF) and found that HGSOC expressing higher levels of vimentin, cMyc, and HE4 in the presence of a low expression level of E-cadherin was carboplatin resistant [38]. It was found that an increased protein expression level of SUSD2 (Notch3-regulating gene) in HGSOC promotes epithelial–mesenchymal transition (EMT), the metastatic capacity of malignant cells, and chemoresistance [39]. Higher expression of oncogene KDM5A was found in ovarian cancer tissues and is associated with cancer cell proliferation, EMT, metastasis, and chemoresistance [40]. Expression of CD44V8-10 (CD44 variant containing exons v8-10) was found associated with chemoresistance and a poor prognosis in ovarian cancer [41].

While there may not be consensus on the exact expression profile of the HGSOC stem-like cells, most in this field believe they exist and may be responsible for tumor relapse and chemoresistance. Henceforth, we will refer to these cells as poorly differentiated stem-like cells (PD-SLCs) (Figure 1). In addition to chemotherapy resistance, the challenge of immune evasion by PD-SLCs within HGSOC tumors has proven to be a large hurdle to overcome as these cells employ mechanisms that reduce immunogenicity such as the altered expression of surface antigens to escape T-cell recognition [42]. However, some of these cells remain susceptible to natural killer (NK) cell-mediated cytotoxicity due to their low MHC-class I expression, allowing NK cells to kill these cells through various mechanisms like the release of cytotoxic molecules, cytokines, or chemokines [43]. This phenomenon has opened a further investigation into using NK cell-based therapies to target PD-SLCs in HGSOC, a malignancy that can harbor tumor cells with the down-modulation of MHC-class I [44].

## 3. Immunosuppression in Ovarian Cancer Patients: Rationale for Increased Metastasis

### 3.1. Immunosuppressive Nature of Ovarian Tumors

To understand the mechanisms of tumor progression, we focused on the role of the tumor cells to mediate immunosuppressive effects within the tumor microenvironment. A prospective cohort study conducted on ovarian cancer patients has shown that increased ascites are associated with a poor prognosis and increased immunosuppression [45]. Intratumoral T cells (T cells with tumor-cell islets) were found to positively contribute to a better prognosis in advanced ovarian carcinoma [46]. Decreased intratumoral T cells were found to be associated with the up-regulation of vascular endothelial growth factor [46]. Ovarian carcinoma ascites-derived tumor-associated T and NK cells were found to exhibit the defective expression and function of signaling proteins including decreased expression levels of TcR-zeta chains and p56 (lck), and reduced gene expression levels of IFN-γ, IL-2, and IL-4 [47]. Although ovarian cancer patients exhibit higher concentrations of NK cells in ascites fluid as compared to peripheral blood, these NK cells lack function [48,49]. Ascites-derived NK cells were found to express low surface expression of CD16, NKp30, and NKG2D, and were less capable of expanding, mediating cytotoxicity, and secreting cytokines in comparison to peripheral blood-derived NK cells [50,51,52]. Lower levels of NK cell numbers were seen in the peripheral blood of ovarian cancer patients [49,53]. The study conducted on the lymphocytes collected from the peritoneal cavity of ovarian cancer patients exhibited dysfunctional lymphocyte-mediated direct and ADCC cytotoxic functions contributing to the spread and proliferation of tumor cells in the peritoneal cavity [48,54].

### 3.2. Markers Associated with Better Prognosis

It was found that the overexpression of the endothelin B receptor (ETBR) is associated with a lack of tumor-infiltrating lymphocytes (TILs) and a shortened life span in ovarian cancer patients [55]. Although CD103+ TILs comprising CD8+ T cells and CD56+ NK cells were found in most ovarian cancer types, they were found to be the most abundant in HGSOCs and were associated with patient survival [56]. CD103 was found to serve as a useful marker to enrich the most beneficial subsets for immunotherapy [56]. Higher percentages, function, and IL-15-mediated activation of ascites-derived NK cells were found to be associated with a better prognosis in ovarian cancer patients [57]. The IL-15 super-agonist complex was found to up-regulate NK cell-mediated cytotoxicity against ovarian cancer cell lines in both in vitro and in vivo studies [58].

### 3.3. Immune Cells Associated with Better Prognosis

A study conducted on the interaction between dendritic cells and T cells has shown that tumor-associated plasmacytoid dendritic cells were found to contribute to tumor immunosuppressive networks in ovarian cancer patients [59]. Tumor-associated T-reg cells were found to be positive for the expression of NKG2D ligand surface expression and were found to be killed by T cells expressing chimeric NKG2D (chNKG2D) receptors due to receptor–ligand interaction in a perforin-dependent manner [60]. Studies have shown that ovarian tumor cells and tumor microenvironmental macrophages produce the chemokine CCL22, responsible for the preferential recruitment and accumulation of T-regs in tumors and in ascites [61]. Blocking the function or migration of T-reg to the tumor microenvironment could fight the battle against ovarian cancer [61]. A higher number of intraepithelial CD8+ TILs and an increased ratio of CD8+ vs. T-reg cells were found to be associated with a better prognosis of epithelial ovarian cancer [62].

### 3.4. Limitations and Challenge in HGSOC Treatments

While NK cell-based immunotherapy holds significant promise for HGSOC treatment, it is essential to acknowledge its potential limitations and challenges. One major challenge is the immunosuppressive tumor microenvironment, which can hinder NK cell function and efficacy. Additionally, the heterogeneity of HGSOC tumors, with varying expression of NK cell-activating ligands and MHC-class I molecules, poses a challenge for targeting all tumor cell populations by primary activated NK cells. 

### 3.5. Future Strategies to Overcome the Challenges of HGSOC Treatments

Future research should focus on developing strategies to overcome these challenges. This includes investigating methods to enhance NK cell infiltration and function within the tumor microenvironment, as well as exploring combination therapies with other immunomodulatory agents or targeted therapies. Additionally, further research is needed to identify and target specific tumor cell populations that may be resistant to primary activated NK cell-mediated cytotoxicity. We have designed and implemented a strategy to use supercharged NK cells as an effective strategy to target the heterogenous nature of ovarian tumors which primary activated NK cells are not capable of targeting.

## 4. Immunotherapies of Ovarian Cancer

Cancer immunotherapies have gained popularity due to their increased effectiveness as a cancer therapy. Humanized monoclonal antibody bevacizumab (anti-vascular endothelial growth factor) adjuvant therapy combined with chemotherapy has been approved for advanced primary and recurrent tumors resulting in a moderate improvement in progression-free survival [63,64,65]. The anti-tumor responses of antibodies against PD1/PDL1 were investigated in ovarian cancer. Monotherapies with nivolumab [66] or pembrolizumab [67] were found to be effective only in a small fraction of ovarian cancer patients. While the combination of nivolumab and ipilimumab therapy was found superior to nivolumab alone, its efficacy remains limited [68]. T cells expressing chimeric NKG2D (chNKG2D) receptors were found to contribute to long-term tumor-free survival in ovarian cancer mouse models [60]. The p53 synthetic long peptide (p53-SLP) was found to be a safe vaccine to induce T-cell responses in ovarian cancer patients [69,70]. Mesothelin-induced CAR-T prolonged survival in an ovarian cancer mouse model [71].

Ovarian tumors have been shown to be targeted by NK cell-mediated killing in several in vitro and in vivo assays [58,72,73,74,75,76], including against 3D ovarian tumor spheroids [75]. It was observed that IP injection of iPSC-derived NK cells and/or peripheral blood-derived NK cells led to high levels of circulating NK cells and the eradication of intraperitoneal ovarian cancer in xenograft mice model [72,73]. Ovarian patient peripheral blood and ascites-derived ex vivo expanded NK cells were found to be cytotoxic against autologous primary ovarian tumor cell lines [76]. Also, allogeneic ex vivo expanded NK cell treatment resulted in a reduced tumor burden and tumor metastasis in mouse models implanted with patient-derived ovarian tumors [77,78]. Adoptive transfer of haploidentical NK cells in combination with chemotherapy was found to be effective in treating ovarian cancer due to the in vivo increased survival and expansion of NK cells; however, strategies to augment in vivo NK cell persistence and expansion need to be explored further [79].

## 5. The Role of MHC-Class I and II Expression on Ovarian Tumors and Their Susceptibility to NK Cell-Mediated Effects

The mechanisms that govern NK activation are still under investigation. However, increased NK cell function has been seen upon the deletion or decrease in many cellular genes in tumors [80]. Most important, the down-modulation of NF-κB and CD44 in tumors induced a significant increase in both cytotoxic activity and the secretion levels of IFN-γ in NK cells in vitro, and induced inflammation and auto-immunity in vivo [14,80,81,82,83]. In our gene knock-down studies, the down-modulation of MHC-class I expression in both transformed and non-transformed healthy cells was found to be one of the underlying mechanisms to induce activation in NK cells [14,80,82,83]. Recent discoveries suggest that NK cell activation is much more complex than we have previously envisioned; it involves many genes/pathways. For example, it was found in mouse studies that DAP10/DAP12 knock-outs resulted in hyper-responsiveness in NK cells [84]. Thus, increased responsiveness of NK cells when key cellular genes were knocked out or knocked down in interacting cells/tumors may point to the fundamental function of NK cells in targeting cells that lose the ability to differentiate optimally, and that the degree of differentiation of the cells is likely the key in regulating NK cell expansion and function.

Using several cancer cell lines, we have previously demonstrated that PDSLCs exhibit lower MHC-class I expression, whereas well-differentiated tumors exhibit much higher expression of surface MHC-class I [14,85,86]. To understand the role of MHC-class I in susceptibility to NK cell-mediated cytotoxicity against ovarian tumors, we have previously used seven ovarian tumor cell lines: OVCAR3 [87,88,89,90], OVCAR4 [91,92,93], OVCAR8 [94,95,96], SKOV3 [77,97,98], Kuramochi [94,99], CaOV3 [77,100,101,102], and OAW28 [86,103,104]. There was a great correlation between the levels of MHC-class I expression and targeting by NK cells. Both OVCAR8 and CAOV3 had minimal expression of MHC-class I and were highly killed by NK cells when compared to the other tumor lines [44]. The down-modulation or loss of MHC-class I and class II has been reported in ovarian carcinoma [105,106,107,108,109] and is one of the factors contributing to escape by the immune system [110]. Increased expression levels of MHC-class I and class II in ovarian cancer correlate with increased numbers of tumor-infiltrating lymphocytes, an increased response to PD1/PDL1 therapy, better prognosis, and the prolonged life of cancer patients [108,109,111,112].

Considering that there is significant down-modulation of surface receptors when cellular genes are decreased or deleted, it is likely that NK cells sense the concentration of receptors on the surface of the cells and react accordingly. The higher the loss of receptors, the more activation of NK cells is expected. In contrast, the rise in surface receptors shuts down the primary NK function mostly known to be due to the down-modulation of MHC-class I. Other receptors are also clearly important in the process. This aspect of NK cell activation requires future investigations.

## 6. Supercharged NK Cells as an Effective Strategy to Eliminate Aggressive Ovarian Tumors

NK cells target tumors via direct cytotoxicity and antibody-dependent cellular cytotoxicity (ADCC) or can indirectly regulate the functions of other immune effectors through their secreted inflammatory cytokines and chemokines [113,114,115]. Increased NK cell activity in peripheral blood and/or NK cell infiltration of tumor tissue are associated with a better prognosis in cancer patients [116,117,118,119]. NK cell-based cancer immunotherapies are found to be effective against several solid tumors [79,120,121,122,123,124]. Studies have shown an increased sensitivity of HGSOC tumor cells to NK cell-mediated cytotoxicity [72,73,79,125] and the ubiquitous expression of NK ligands on primary HGSOC tumors [32]. Our data also reveal higher NK-mediated cytotoxicity against HGSOC cell lines expressing lower MHC-class I levels [44]. Despite the potential of NK cells in targeting HGSOC, their effectiveness is limited by several factors: (i) a restricted number of NK cells in the tumor, (ii) dysfunctional endogenous NK cells among ovarian cancer patients, (iii) the short lifespan of NK cells in vivo, (iv) difficulties in NK cell infiltration to the tumor site, particularly in solid tumors, and (v) functional impairments in the tumor microenvironment [43,126,127].

To overcome the challenges mentioned above regarding NK-cell therapies, our laboratory has developed the technology to up-regulate the proliferative, effector, and cytotoxic function of NK cells. This technique involves the activation of peripheral blood-derived NK cells with a combination of IL-2 and anti-CD16 mAbs, and co-culturing of activated NK cells with osteoclasts (OCs) as feeder cells in the presence of probiotics sAJ2, this process leads to expansion and highly functional activation of NK cells [103,128,129]. Due to their superior anti-cancer activity, these cells were named supercharged NK (sNK) cells [15,16,86,103,128,129,130,131,132,133,134,135,136] (Table 1 and Figure 2). sNK cells were found to be highly effective against tumors in preclinical models [15,16,86,103,128,129,130,131,132,133,134,135,136]. The rationale for using OCs for this methodology was the secretion of a wide range of cytokines and chemokines including IL-12, IL-15, IFN-a, and IL-18 by OCs, which are known to increase NK function. In addition, OCs express the important NK-activating ligands MICA/B and ULBPs [137,138]. The probiotic sAJ2 is a combination of Gram-positive probiotic bacteria strains: *Streptococcus thermophiles, Bifidobacterium longum*, *Bifidobacterium breve*, *Bifidobacterium infantis*, *Lactobacillus acidophilus*, *Lactobacillus plantarum*, *Lactobacillus casei*, and *Lactobacillus bulgaricus*. The rationale for using probiotic sAJ2 treatments was to increase the cytokines secreted by NK cells including IFN-γ which could facilitate the signals required for NK cell expansion [85,86,128,135,139]. Therefore, combining both probiotics and OCs led to the induction of signals participating in the expansion, survival, and functional activation of NK cells (Table 1 and Figure 2).

We have previously demonstrated that sNK cells exhibit a significantly higher capability for cytokine secretion and cytotoxic function against several cancers (including ovarian cancers) in comparison to activated primary NK cells [128,130,140]. Both the cytokine secretion and cytotoxic activity of sNK cells were maintained for over 30 days (donor dependent: average 27–36 days), as opposed to the 5–7-day duration observed in primary activated NK cells [128,130,140]. Thus far, it is very well established that primary IL-2-activated NK cells typically target PDSLCs and exhibit a slight/low killing potential against well-differentiated tumors. This pattern did not apply to the sNK-induced killing of tumors because sNK cells were found to target both PDSLCs and well-differentiated tumors. We observed that sNK cells induced a significantly greater killing of differentiated tumors when compared to primary activated NK cells [44,128,129,132,135,140]. In addition, IFN-γ secreted by sNK cells induced significantly higher differentiation of PDSLCs when compared to primary activated NK cells [128,130] (Table 1). Although differentiated tumors become resistant to primary activated NK cell-mediated cytotoxicity, they remain susceptible to sNK cell-mediated cytotoxicity and chemotherapeutic drugs [14,130,132], and such an effect was also seen in ovarian cancer cell lines [44].

In our recent study, we found that sNK cells were the only NK cells that effectively lysed both well-differentiated and poorly differentiated oral and ovarian tumors, as evidenced in both the normalized cell index and at the level of cytolysis, whereas primary activated NK cells could only lyse poorly differentiated tumors. In all cases, sNK cells greatly lysed the tumor cells and no visible tumor could be visualized. Therefore, sNK cells behave not only as NK cells but also as T cells, since they are capable of lysing MHC-class I-bearing well-differentiated tumors.

We are now in the process of investigating the efficacy of sNK cells in ovarian tumor-bearing humanized-BLT mice. In our previous studies, we performed one intravenous injection of sNK cells in oral [130,135], pancreatic [86,130], and melanoma [manuscript in prep] tumors implanted humanized-BLT mice, and mice were monitored for 4–5 weeks. Mice received sNK cells therapy had an increased lifespan and decreased tumor mass in comparison to mice receiving no treatment [86,130,135]. Tumors dissected from the sNK cell-treated group expressed higher expression levels of human CD45, MHC-class I, CD54, and B7H1 [86,130,135]. These results suggest that sNK cells increased tumor immune cell infiltration and induced in vivo tumor differentiation. In addition, the restored cytotoxic function and cytokine secretion of immune cells isolated from the spleen, peripheral blood, gingiva, pancreas, and bone marrow of sNK cell-treated tumor-bearing hu-BLT mice were observed [86,130,135].

## 7. Clinical Trials of NK Cell-Based Ovarian Cancer Therapeutics

Several immunotherapeutic clinical trials have demonstrated the efficacy of immunotherapy as a treatment modality for cancers [141,142,143,144,145]. NK cell-based immunotherapies have demonstrated benefits in clinical trials to treat several cancers including ovarian cancer [123,146,147,148,149,150]. Allogeneic NK cell transfer is safe and feasible because NK cells do not require HLA matching [151]. Geller et al. (ClinicalTrials.gov. identifier: NCT01105650) enrolled 14 ovarian cancer patients, which were infused with haploidentical IL-2 activated NK cells followed by SC IL-2 infusion three times per week for fourteen days, after which NK cell expansion was detected in the peripheral blood of patients [79]. Xie et al. documented a case report of ovarian cancer patient receiving ex vivo expanded NK cells every two weeks for a total of six infusions; this approach resulted in a significant reduction in tumor mass and prolonged patient survival in the presence of minimal adverse effects [78]. A summary of NK cell-based clinical trials for ovarian cancer is listed in Table 2. In total, 6 out of 14 patients enrolled in the clinical trial NCT01105650 were found to have progression-free survival at one year. For other listed trials, evaluations such as safety, progression-free survival, or overall survival are still under investigation.

In addition, we have recently initiated clinical trials using sNK cells. The results from our first two patients exhibited an unparalleled safety profile and initial positive results. We will be continuously monitoring these patients to obtain data for their efficacy.

## 8. Conclusions

Both primary activated NK cells and sNK cells play a crucial role in the direct killing and differentiation of ovarian tumors. However, sNK cells are more potent, ultimately preventing tumor establishment by the specific recognition, selection, and differentiation of tumors. Preclinical investigations and clinical trials (ongoing and completed) have demonstrated the promising anti-cancer effects of primary activated NK cell-based immunotherapy. Because sNK cells induce significant direct killing and/or differentiation of PDSLCs, and chemotherapy selectively targets differentiated tumors, a combination of both sNK and chemotherapy could be the most effective strategy to eliminate the heterogeneous population of ovarian tumors (Figure 3). Current work in progress will provide further information needed to establish efficacy based on the tumor site, stage of cancer, and patient-related factors, and add more information regarding the optimal dosage of NK cell infusion in combination with other therapeutics. The work presented in this review suggests NK cell-based therapy will be an important part of the armamentarium of ovarian cancer therapeutics.

## Figures and Tables

**Figure 1 vaccines-12-00677-f001:**
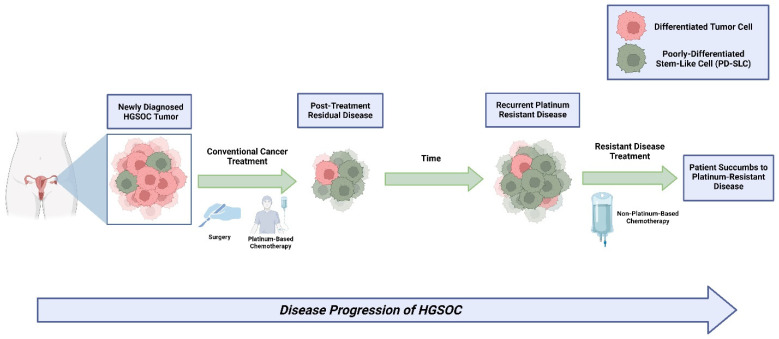
Schematic presentation of disease progression of HGSOC: HGSOCs likely consist of PDSLCs and differentiated tumor cells. Conventional therapeutics are incapable of completely eradicating PDSLCs contributing to disease progression and tumor relapse.

**Figure 2 vaccines-12-00677-f002:**
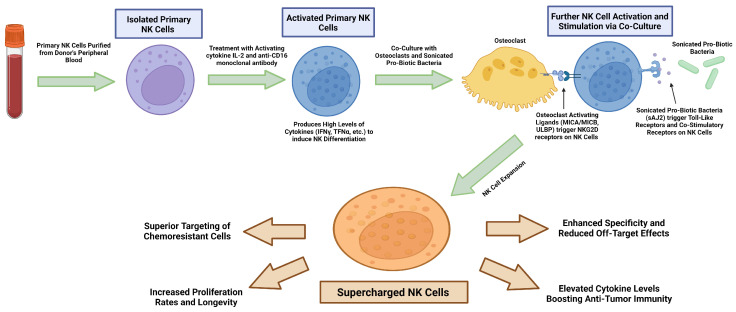
Schematic presentation of generation and characteristics of supercharging NK cells. NK cells (NK cells (0.5 × 10^6^ cells/2 mL)) are purified from the peripheral blood of human donors and are treated with a combination of IL-2 (1000 U/mL) and anti-CD16mAb (3 µg/mL) overnight. Activated NK cells are then co-cultured with osteoclasts and probiotic bacteria (1:2:4: osteoclasts: NK: PB). Media were refreshed every three days for an average of 27–36 days. Supercharged NK cells exhibit significantly increased functional activity in comparison to primary NK cells.

**Figure 3 vaccines-12-00677-f003:**
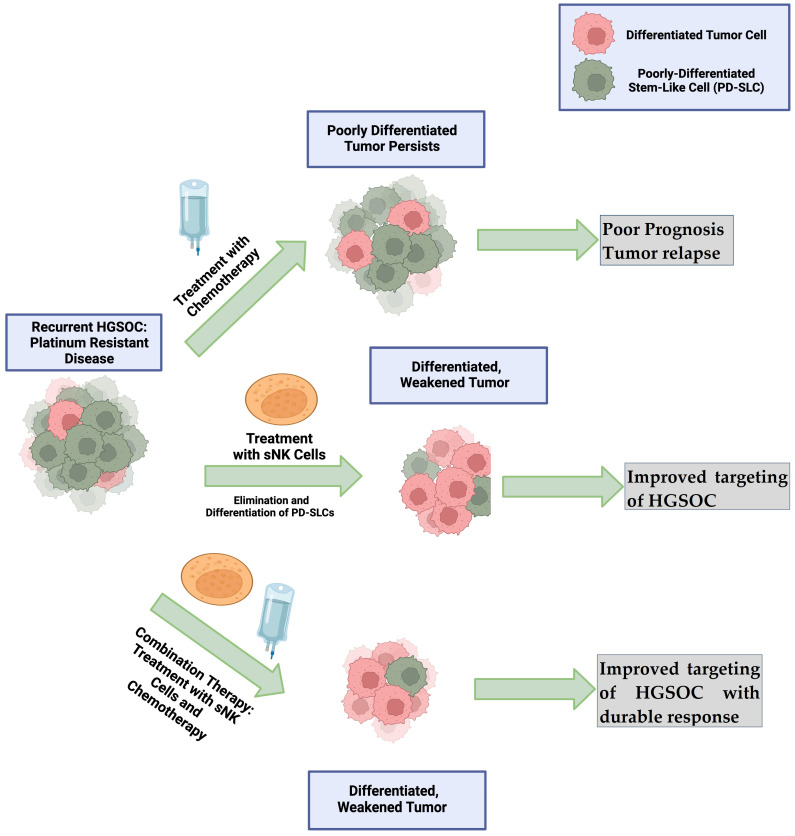
Hypothetical presentation of effective therapeutics for ovarian cancer. Chemotherapy alone could target well-differentiated tumors, but not PDSCSs, which could result in tumor recurrence. sNK cells induce killing against both PDSLCs and well-differentiated tumors and induce the differentiation of PDSLCs. A combination of sNK and chemotherapy could be the most effective strategy to eliminate the heterogeneous population of ovarian tumors: sNK therapies could help to kill both PDSLCs and well-differentiated tumors and induce the differentiation of PDSLCs, which could then allow chemotherapy to clean well-differentiated and NK-induced differentiated tumors.

**Table 1 vaccines-12-00677-t001:** Characteristic differences of primary vs. super-charged NK cells.

	Primary IL-2-Activated NK Cells	Supercharged NK Cells	References
Lysis of PDSLCs	++	+++++ (****)	[86,128,130]
Effector function associated gene-expression: IRF1, JUN, STAT1, H1F1A	++	+++ (***)	[130]
ADCC against tumors	++	+++ (**)	[132]
Cell survival	6–9 days	27–36 days	[130]
Cell survival associated gene-expression: STAT2, IRF9	+	+++ (***)	[130]
Cell expansion	+/−	++++ (****)	[130]
Protein expression of cytotoxic granules	+	++ (**)	[130]
Activating receptors surface expression	++	++++ (****)	[130]
Inhibitory receptors surface expression levels	+++ (***)	+	[130]
Stage of cell cycle	% of cells: G1 > S > G2M	% of cells: G2M > S > G1	[130]
	Majority of cells in sNK are highly proliferative		
NK supernatant mediated tumor differentiation	++	++++ (****)	[86,128,130]
Selection and expansion of CD8+ T cells (in vivo and in vitro)	+	+++ (***)	[129,135]
In vivo tumor growth and metastasis inhibition	+	++++ (****)	[86,135]

PDSLCs: poorly-differentiated-cancer-stem-like-cells; ADCC: antibody-dependent cell cytotoxicity; Cytotoxic granules: granzyme B, cathepsin C, and perforin-1; + (1–2 fold), ++ (2–4 fold), +++ (3–4 fold), ++++ (4–6 fold), +++++ (>6 fold). **** (*p* value < 0.0001), *** (*p* value < 0.001), ** (*p* value 0.001–0.01).

**Table 2 vaccines-12-00677-t002:** Clinical trial of NK cell-based ovarian cancer immunotherapy.

Treatment Approach	Clinical Trial Stage	ClinicalTrials.gov Identifier
Allogeneic IV NK + IL-2 (IV)	Completed	NCT01105650
Haploidentical NK + IL-2 + indoleamine-2,3-dioygenas (IDO) (IP)	Completed	NCT02118285
Allogeneic NK + IL-2 (IP)	Completed	NCT03213964
Cryosurgery + NK	Completed	NCT02849353
TROP2-CAR IL-15 Transduced CB-NK (IP)	In progress	NCT05922930
Anti-mesothelin CAR-NK	In progress	NCT03692637
Autologous activated NK (IV)	In progress	NCT03634501
Ex vivo generated UCB-derived allogeneic NK + IL-2 (IP)	In progress	NCT03539406
Cytokine-induced NK cells + radiofrequency ablation	In progress	NCT02487693

UCB: umbilical cord blood-derived; IV: intravenous; IP: intraperitoneal; CB: cord blood.

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
