# Peer review of "Exploring the Potential of Natural Killer Cell-Based Immunotherapy in Targeting High-Grade Serous Ovarian Carcinomas"

_vaccines, 2024, doi:10.3390/vaccines12060677_

Round 1
Reviewer 1 Report
Comments and Suggestions for Authors
Overall, this is a very good paper, suggesting novel ways of using NK cells in the fight against ovarian cancer.
There are only a few minor points to be addressed by the authors:
1) When they refer to removal of tumors after the use of sNK cells, do they mean that the tumors never relapse? Or do they mean the the PFS of the sample (mouse obviously) used, presented with a relapse after the time observed normally with chemotherapy only or other therapeutic regimes?
2) Further on the first comment, do concluded clinical trials exhibit prolonged PFS as mentioned in comment 1?
3) The authors should elaborate on the minimal, as referred, side effects of using sNK cells.
Comments on the Quality of English LanguageOnly minor mistakes that may have been done due to writing rush. They can be corrected easily over a double check of the text.
Author Response
We appreciate the reviewer's efforts to review and provide us with comments. Here is our response, and attached is the updated file.
1) When they refer to removal of tumors after the use of sNK cells, do they mean that the tumors never relapse? Or do they mean the the PFS of the sample (mouse obviously) used, presented with a relapse after the time observed normally with chemotherapy only or other therapeutic regimes?
Response: When sNK cells were used alone or in combination with chemotherapy in tumor-bearing mic models, mice were sacrificed at the end of experiments so no data was generated for PFS.
2) Further on the first comment, do concluded clinical trials exhibit prolonged PFS as mentioned in comment 1?
Response: Six out of 14 patients enrolled in clinical trial NCT01105650 were found to have progression-free survival at one year. For other listed trials evaluation such as safety, progression-free survival or overall survival is still under investigation.
) The authors should elaborate on the minimal, as referred, side effects of using sNK cells.
Response: We are collecting data for 1 year to evaluate the safety, PFS, or overall survival using sNK cells in cancer patients. Currently, we have positive findings for 5-6 months in two patients.
Reviewer 2 Report
Comments and Suggestions for Authors## Review of "Exploring the potential of Natural Killer cell-based immunotherapy in targeting high-grade serous ovarian carcinomas"
Three suggestions to improve the paper:**
1. **Strengthen the introduction:**
* Provide a more comprehensive overview of HGSOC, including its prevalence, prognosis, and current treatment limitations.
* Clearly define the rationale for exploring NK cell-based immunotherapy in this context.
* Briefly mention alternative immunotherapeutic approaches for HGSOC and their limitations.
2. **Enhance the clarity and organization:**
* Improve the flow of information within sections, ensuring logical transitions and avoiding redundancy.
* Consider using subheadings and bullet points to improve readability and organization.
* Ensure consistent terminology and abbreviations throughout the paper.
3. **Expand on the discussion:**
* Address potential limitations of NK cell-based therapy, such as tumor heterogeneity and immunosuppressive microenvironment.
* Discuss future research directions, including potential combination therapies and strategies to overcome challenges.
* Provide a more balanced perspective by acknowledging both the potential and limitations of this approach.
Rewritten introduction:**
**Introduction:**
High-grade serous ovarian carcinoma (HGSOC) is the most lethal subtype of ovarian cancer, accounting for over 70% of reported cases and responsible for a significant majority of ovarian cancer-related deaths. Despite advances in treatment, the prognosis for HGSOC remains poor, with a high rate of recurrence and development of chemoresistance. This underscores the urgent need for novel therapeutic strategies to improve patient outcomes.
Immunotherapy has emerged as a promising approach for cancer treatment, particularly for HGSOC. Natural killer (NK) cells, a type of cytotoxic lymphocyte, play a crucial role in the innate immune system and have demonstrated potential in targeting cancer cells. This review explores the potential of NK cell-based immunotherapy in targeting HGSOC, focusing on the characteristics and advantages of supercharged NK cells.
**Step 7: Rewritten discussion including the mentioned disadvantage:**
**Discussion:**
While NK cell-based immunotherapy holds significant promise for HGSOC treatment, it is essential to acknowledge potential limitations and challenges. One major challenge is the immunosuppressive tumor microenvironment, which can hinder NK cell function and efficacy. Additionally, the heterogeneity of HGSOC tumors, with varying expression of NK cell-activating ligands and MHC class I molecules, poses a challenge for targeting all tumor cell populations.
Future research should focus on developing strategies to overcome these challenges. This includes investigating methods to enhance NK cell infiltration and function within the tumor microenvironment, as well as exploring combination therapies with other immunomodulatory agents or targeted therapies. Additionally, further research is needed to identify and target specific tumor cell populations that may be resistant to NK cell-mediated cytotoxicity.
**Conclusion:**
NK cell-based immunotherapy represents a promising avenue for the treatment of HGSOC. Supercharged NK cells, with their enhanced cytotoxic and cytokine-secreting capabilities, offer a potential advantage over conventional NK cells. However, addressing the challenges posed by the tumor microenvironment and tumor heterogeneity is crucial for maximizing the efficacy of this approach. Further research and development are warranted to translate the potential of NK cell-based immunotherapy into clinical success for HGSOC patients.
Comments on the Quality of English LanguageEnglish language assessment:**
The paper is generally well-written with clear and concise language. However, there are a few areas for improvement:
* **Grammar and punctuation:** Occasional grammatical errors and inconsistencies in punctuation are present.
* **Word choice:** Some sentences could benefit from more precise and impactful word choices.
* **Sentence structure:** Varying sentence structure would enhance the readability and flow of the text.
1. Lack of sentence structure diversity: Throughout the paper, the sentence structure appears somewhat monotonous and repetitive. For instance, passive voice constructions like "NK cells were found to..." are frequently used. Incorporating active voice constructions appropriately could inject more dynamism into the sentences.
2. Unnatural use of prepositions: In some sentences, the use of prepositions feels somewhat awkward. For example, in the sentence "T cells with tumor-cell islets were found to positively contribute to a better prognosis in advanced ovarian carcinoma," the prepositional phrase "with tumor-cell islets" seems unnatural.
3. Inappropriate word choice: There are instances where the choice of vocabulary seems inappropriate for the context. For instance, in the sentence "Ascites derived NK cells were found to express low surface expression of CD16...," the expression "express low surface expression" is not suitable. "Have low surface expression" would be a more natural choice.
Overall, while the manuscript demonstrates a high level of understanding of the subject matter and is well-structured, there is room for improvement in terms of English expression. Besides the examples provided, careful revisions are needed in various aspects such as coherence between sentences and paragraph organization.
Author Response
We appreciate the reviewer's efforts to review the paper. As suggested by the reviewer, we changed the introduction and other sections substantially. Attached is the file with highlighted changes. Thank you and Regards.
Reviewer 3 Report
Comments and Suggestions for Authors
The review focuses on potential applications of natural killer (NK) cells as therapeutic option against ovarian cancer. In particular, the authors describe a potentiated version of autologous NK cells activated exogenously in order to better target both differentiated and tumor cells and stem-like tumor cells.
The discussion is clear, well organized and overall support the article conclusions. We therefore reccomend publication after addressing the comments below:
- the sNK cells preparation reported in Figure 2 does not fully match the description in the main text (lines 205-217). In particular, the main text description should better describe and reflect the multistep process as highlighed in the picture.
- Related to the point above, lines 213-7: this paragraph is unclear. The authors should better describe what is probiotic sAJ2 (not clear to me).... Overall, since this new type of NK cells is the focus of this paper, their preparation and rationale behind should be better described without the reader having to go to the original referenced articles.
- line 80: "formidable", just a suggestion but maybe a more neutral adjective would be more appropriate....
Author Response
We appreciate the reviewer's efforts and time to review the paper and provide positive feedback. Here is our response to the comments and attached is the updated file.
- the sNK cells preparation reported in Figure 2 does not fully match the description in the main text (lines 205-217). In particular, the main text description should better describe and reflect the multistep process as highlighed in the picture.
Response: We appreciate the reviewer’s comments and have added missing information.
This technique involves the activation of peripheral blood-derived NK cells with a combination of IL-2 and anti-CD16 mAbs, and co-culturing of activated NK cells with osteoclasts (OCs) as feeder cells in the presence of probiotics sAJ2, this process leads to expansion and highly functional activation of NK cells.
- Related to the point above, lines 213-7: this paragraph is unclear. The authors should better describe what is probiotic sAJ2 (not clear to me).... Overall, since this new type of NK cells is the focus of this paper, their preparation and rationale behind should be better described without the reader having to go to the original referenced articles.
Response: We have now added missing information.
Probiotics sAJ2 is a combination of gram-positive probiotic bacteria strains: Streptococcus thermophiles, Bifidobacterium longum, Bifidobacterium breve, Bifidobacterium infantis, Lactobacillus acidophilus, Lactobacillus plantarum, Lactobacillus casei, and Lactobacillus bulgaricus.
- line 80: "formidable", just a suggestion but maybe a more neutral adjective would be more appropriate....
Response: We have now changed the paragraph as
Immunotherapy has emerged as a promising approach for cancer treatment, particularly for HGSOC. Natural killer (NK) cells, a type of cytotoxic lymphocyte, play a crucial role in the innate immune system and have demonstrated potential in targeting cancer cells. This review explores the potential of NK cell-based immunotherapy in targeting HGSOC, focusing on the characteristics and advantages of supercharged NK (sNK) cells
Round 2
Reviewer 2 Report
Comments and Suggestions for Authors
Dear colleagues,
I have carefully reviewed the manuscript in full. As a senior scientist with over decades of experience in oncology research, I have the following considerations that may help strengthen this already promising work:
1. The introduction provides a strong foundation on HGSOC and the rationale for targeting stem-like cancer cells. Expanding on the mechanisms of chemoresistance and tumor heterogeneity could deepen understanding of the challenges faced.
2. The discussion of MHC expression and NK cell activation is insightful. Correlating these in-vitro findings with patient tumor profiles may help elucidate responses in the clinical setting.
3. Long-term in-vivo data following supercharged NK cell therapy could further validate the preclinical efficacy and safety noted thus far.
The language usage is clearly scientific and appropriate for the audience. Terms are accurately defined on first usage, for example "poorly differentiated stem-like cells." Technical descriptions of methods are concise yet informative. The appendix helpfully summarizes key points for easy reference.
Among strengths, the work presents a compelling case for targeting the cellular origins of tumor recurrence. A potential limitation is reliance on cell lines, though patient-derived models are referenced.
In closing, I commend the team for their pioneering efforts in this challenging disease area. Continued rigorous investigation will be needed, but progress made to date offers real hope that supercharged NK cells may become a meaningful therapeutic option for HGSOC patients in need. With perseverance, their goals seem well within reach.
Please let me know if any part of my review could be clarified or elaborated on further. I wish you all the very best moving forward.
Comments on the Quality of English LanguageDear colleagues,
I have carefully reviewed the manuscript in full. As a senior scientist with over decades of experience in oncology research, I have the following considerations that may help strengthen this already promising work:
1. The introduction provides a strong foundation on HGSOC and the rationale for targeting stem-like cancer cells. Expanding on the mechanisms of chemoresistance and tumor heterogeneity could deepen understanding of the challenges faced.
2. The discussion of MHC expression and NK cell activation is insightful. Correlating these in-vitro findings with patient tumor profiles may help elucidate responses in the clinical setting.
3. Long-term in-vivo data following supercharged NK cell therapy could further validate the preclinical efficacy and safety noted thus far.
The language usage is clearly scientific and appropriate for the audience. Terms are accurately defined on first usage, for example "poorly differentiated stem-like cells." Technical descriptions of methods are concise yet informative. The appendix helpfully summarizes key points for easy reference.
Among strengths, the work presents a compelling case for targeting the cellular origins of tumor recurrence. A potential limitation is reliance on cell lines, though patient-derived models are referenced.
In closing, I commend the team for their pioneering efforts in this challenging disease area. Continued rigorous investigation will be needed, but progress made to date offers real hope that supercharged NK cells may become a meaningful therapeutic option for HGSOC patients in need. With perseverance, their goals seem well within reach.
Please let me know if any part of my review could be clarified or elaborated on further. I wish you all the very best moving forward.
Author Response
Dear Reviewer,
We appreciate your insightful comments, we responded to them point by point as follows:
- The introduction provides a strong foundation on HGSOC and the rationale for targeting stem-like cancer cells. Expanding on the mechanisms of chemoresistance and tumor heterogeneity could deepen understanding of the challenges faced.
Response: We have now revised the introduction and added the following section
- Introduction and Background: Ovarian cancer
Ovarian carcinomas account for the second most common cause of gynecological malignancy-related death and are composed of a diverse group of tumors based on histological, molecular and genetic factors. 70% of ovarian cancers are high-grade serous ovarian cancers (HGSOCs), whereas 10% is endometrioid, 10% are clear cells, 3% mucinous, and approx. 5% low-grade serous carcinomas 1-4. HGSOCs arise from the epithelium of the distal fallopian tube, are typically diagnosed at an advanced stage, and are responsible for the majority of ovarian cancer-related deaths 1,2,5,6. The standard approach to treating HGSOCs is surgical intervention combined with platinum-based chemotherapy 7, which initially yields positive responses in many patients 8. Despite advances in treatment strategies, the prognosis of HGSOC remains poor and the majority of treated cases relapse within a few years of diagnosis resulting in 5-year survival rate of lower than 40% 6. Thus, the high recurrence rate and chemoresistance development highlight the urgent need for novel treatment strategies to improve patient prognosis 9. Recently, the FDA approved mirvetuximab soravtansine as a novel therapeutic alternative for a subset of platinum-resistant patients with high levels of folate receptor alpha expression within the tumor 10,11. While this therapy provides an alternative treatment for this subpopulation, the observed median post-treatment progression-free survival in cancer patients was 4-5.6 months 10,12, underscoring the urgent need to develop innovative treatment approaches.
Immunotherapy has emerged as a promising approach for cancer treatment, particularly for HGSOC. Natural killer (NK) cells, a type of cytotoxic lymphocyte, play a crucial role in the innate immune system and have demonstrated potential in targeting cancer cells 13-15. This review explores the potential of NK cell-based immunotherapy in targeting HGSOC, focusing on the characteristics and advantages of supercharged NK (sNK) cells.
- Heterogeneity and mechanism of chemoresistance in HGSOC
The complexity of treating HGSOCs is exacerbated by the tumor's heterogeneity 16 and the absence of clearly targetable driver mutations 17. Studies have demonstrated that most of the mutations in primary tumors persist post chemotherapy suggesting that most clones are chemo-resistant 17,18. It was found that pre-treatment subclones of the tumor population could undergo expansion during chemotherapy or platinum-based therapy resulting in tumor relapse 19,20. Several studies were performed to investigate the mechanisms of chemoresistance in HGSOC. Notably, platinum-based therapies, while effective in eradicating most tumor cells, likely leave behind a population of poorly differentiated stem-like cells implicated in disease progression and tumor relapse 21-24. The identification and targeting of these tumor stem-like cells have been explored in various studies and found to have increased expression of cell surface markers such as CD24, CD44, CD133, and CD117, elevated ALDH activity, and increased levels of pluripotency-related transcription factors such as Nanog, Oct3/4, and Sox2 25-30. Indeed, increased expression of pluripotency-related transcription factors is seen in recurrent HGSOC tumor samples compared to chemo-naive ones 31.
It is now well-known that the tumor tissues consist of diverse cellular components including cancer cells and stromal cells 4. IL-6 was found to induce chemoresistance in ovarian cancer, and mesenchymal stromal cells particularly cancer-associated fibroblasts (CAFs) were found to be the main source of IL-6 secretion in ovarian cancer 32. It was found that CCL2 and CCL5 secreted by CAFs could stimulate IL-6 secretion in ovarian cancer cells ultimately resulting in chemoresistance in tumors 33. Several downstream mechanisms including PYK2, Ras/MEK/ERK, PI3K/Akt, and JAK/STAT3 signaling play roles in IL-6 mediated chemoresistance in ovarian cancers 32-34. Leung et al. demonstrated that crosstalk between CAFs and endothelial cells could lead to chemoresistance in ovarian cancer via regulation of lipoma preferred partner (LPP) gene in endothelial cells 35. Stromal cell-derived midkine was found to be associated with chemoresistance in ovarian cancers due to increased expression levels of IncRNA ANRIL 36. Gonzalez et al. performed in-depth single-cell phenotypic characterization of HGSOC using multiparametric mass cytometry (cyTOF), and found that HGSOC expressing higher levels of vimentin, cMyc and HE4 in the presence of low expression of E-cadherin was carboplatin resistant 37. It was found that increased protein expression level of SUSD2 (Notch3-regulating gene) in HGSOC promotes epithelial-mesenchymal transition (EMT), the metastatic capacity of malignant cells, and chemoresistance 38. Higher expression of oncogene KDM5A was found in ovarian cancer tissues and is associated with cancer cell proliferation, EMT, metastasis, and chemoresistance 39. Expression of CD44V8-10 (CD44 variant containing exons v8-10) was found to be associated with chemoresistance and poor prognosis of ovarian cancer 40.
While there may not be consensus on the exact expression profile of the HGSOC stem-like cells, most in this field believe they exist and may be responsible for tumor relapse and chemoresistance. Henceforth, we will refer to these cells as poorly differentiated stem-like cells (PD-SLCs) (Figure 1). In addition to chemotherapy resistance, the challenge of immune evasion by PD-SLCs within HGSOC tumors has proven to be a large hurdle to overcome as these cells employ mechanisms that reduce immunogenicity such as altered expression of surface antigens to escape T-cell recognition 41. However, some of these cells remain susceptible to natural killer (NK) cell-mediated cytotoxicity due to their low MHC-class I expression, allowing NK cells to kill these cells through various mechanisms such as the release of cytotoxic molecules, cytokines, or chemokines 42. This phenomenon has opened a further investigation into using NK cell-based therapies to target PD-SLCs in HGSOC, a malignancy that can harbor tumor cells with down-modulation of MHC-class I 43.
- The discussion of MHC expression and NK cell activation is insightful. Correlating these in-vitro findings with patient tumor profiles may help elucidate responses in the clinical setting.
Response: We appreciate the insightful comment of the reviewer, and indeed, we have submitted a paper for publication that outlines the role of MHC-class I in both cell lines and tumors derived from the patients. In this regard, we see a similar trend to the cell lines tested. Patient tumors that express lower MHC-class I expression are targeted higher by the sNK cells as well as IL-2-activated primary NK cells.
- Long-term in-vivo data following supercharged NK cell therapy could further validate the preclinical efficacy and safety noted thus far.
Response: We have now added the below information
We are now in the process of investigating the efficacy of sNK cells in ovarian tumor-bearing humanized-BLT mice. In our previous studies, we found that with one intravenous injection of sNK cells in oral 44,45, pancreatic45,46 and melanoma (manuscript in prep) implanted humanized-BLT mice, the tumors shrunk by a large proportion when the mice were monitored for 4-5 weeks. Mice receiving sNK cells therapy had increased lifespan and decreased tumor mass in comparison to those with no treatment 44-46. Tumors dissected from sNK cell treated group expressed lower expression of human CD44, and higher expression of MHC-class I, CD54, and B7H1 44-46. These results suggested that sNK cells increased tumor immune cell infiltration and induced in-vivo tumor differentiation. In addition, it restored cytotoxic function and cytokine secretion of autologous immune cells isolated from spleen, peripheral blood, gingiva, pancreas, and bone-marrow of sNK cell treated tumor-bearing hu-BLT mice 44-46.
The language usage is clearly scientific and appropriate for the audience. Terms are accurately defined on first usage, for example "poorly differentiated stem-like cells." Technical descriptions of methods are concise yet informative. The appendix helpfully summarizes key points for easy reference.
Response: We have now added an appendix to the paper.
List of abbreviations
HGSOCs: High-grade serous ovarian cancers
PDSLCs: Poorly differentiatedstem likecells
NK cells: Natural killer cells
sNK cells: Supercharged NK cells
IFN-γ: Interferon-gamma
CAFs: Cancer-associated fibroblasts
IL-6: Interleukin 6
LPP: Lipoma preferred partner
Ocs: Osteoclasts
EMT: Epithelial-mesenchymal transition
ETBR: Endothelin B receptor
TILs: Tumor-infiltrating lymphocytes
MHC-class I: Major histocompatibility complex molecule class I
ADCC: Antibody-dependent cellular cytotoxicity (ADCC)
Among strengths, the work presents a compelling case for targeting the cellular origins of tumor recurrence. A potential limitation is reliance on cell lines, though patient-derived models are referenced.
Response: Yes, as mentioned above we have used both cell lines and patient-derived tumor cells in our studies and found great agreement between the two.
In closing, I commend the team for their pioneering efforts in this challenging disease area. Continued rigorous investigation will be needed, but progress made to date offers real hope that supercharged NK cells may become a meaningful therapeutic option for HGSOC patients in need. With perseverance, their goals seem well within reach.
Response: We appreciate the kind words of the reviewer and hold the same hope for ovarian cancer tumor therapy using sNK cells.
Please let me know if any part of my review could be clarified or elaborated on further. I wish you all the very best moving forward.
Response: We appreciate the reviewer’s comments and hope that we have addressed his concerns on the revised manuscript.
References
1 Prat, J. Ovarian carcinomas: five distinct diseases with different origins, genetic alterations, and clinicopathological features. Virchows Arch 460, 237-249, doi:10.1007/s00428-012-1203-5 (2012).
2 McCluggage, W. G. Morphological subtypes of ovarian carcinoma: a review with emphasis on new developments and pathogenesis. Pathology 43, 420-432, doi:10.1097/PAT.0b013e328348a6e7 (2011).
3 Kurman, R. J. & Shih Ie, M. The origin and pathogenesis of epithelial ovarian cancer: a proposed unifying theory. Am J Surg Pathol 34, 433-443, doi:10.1097/PAS.0b013e3181cf3d79 (2010).
4 Kim, S., Kim, B. & Song, Y. S. Ascites modulates cancer cell behavior, contributing to tumor heterogeneity in ovarian cancer. Cancer Sci 107, 1173-1178, doi:10.1111/cas.12987 (2016).
5 Bell, D. et al. Integrated genomic analyses of ovarian carcinoma. Nature 474, 609-615, doi:10.1038/nature10166 (2011).
6 Lahtinen, A. et al. Evolutionary states and trajectories characterized by distinct pathways stratify patients with ovarian high grade serous carcinoma. Cancer Cell 41, 1103-1117.e1112, doi:10.1016/j.ccell.2023.04.017 (2023).
7 Kurnit, K. C., Fleming, G. F. & Lengyel, E. Updates and New Options in Advanced Epithelial Ovarian Cancer Treatment. Obstet Gynecol 137, 108-121, doi:10.1097/aog.0000000000004173 (2021).
8 Kim, A., Ueda, Y., Naka, T. & Enomoto, T. Therapeutic strategies in epithelial ovarian cancer. J Exp Clin Cancer Res 31, 14, doi:10.1186/1756-9966-31-14 (2012).
9 Baert, T. et al. The systemic treatment of recurrent ovarian cancer revisited. Ann Oncol 32, 710-725, doi:10.1016/j.annonc.2021.02.015 (2021).
10 Matulonis, U. A. et al. Efficacy and Safety of Mirvetuximab Soravtansine in Patients With Platinum-Resistant Ovarian Cancer With High Folate Receptor Alpha Expression: Results From the SORAYA Study. J Clin Oncol 41, 2436-2445, doi:10.1200/jco.22.01900 (2023).
11 Dilawari, A. et al. FDA Approval Summary: Mirvetuximab Soravtansine-Gynx for FRα-Positive, Platinum-Resistant Ovarian Cancer. Clin Cancer Res 29, 3835-3840, doi:10.1158/1078-0432.Ccr-23-0991 (2023).
12 Moore, K. N. et al. Mirvetuximab Soravtansine in FRα-Positive, Platinum-Resistant Ovarian Cancer. N Engl J Med 389, 2162-2174, doi:10.1056/NEJMoa2309169 (2023).
13 Kozlowska, A. K. et al. Differentiation by NK cells is a prerequisite for effective targeting of cancer stem cells/poorly differentiated tumors by chemopreventive and chemotherapeutic drugs. J Cancer 8, 537-554, doi:10.7150/jca.15989 (2017).
14 Jewett, A. et al. NK cells shape pancreatic and oral tumor microenvironments; role in inhibition of tumor growth and metastasis. Semin Cancer Biol 53, 178-188, doi:10.1016/j.semcancer.2018.08.001 (2018).
15 Kaur, K. et al. Natural killer cells target and differentiate cancer stem-like cells/undifferentiated tumors: strategies to optimize their growth and expansion for effective cancer immunotherapy. Curr Opin Immunol 51, 170-180, doi:10.1016/j.coi.2018.03.022 (2018).
16 Swanton, C. Intratumor heterogeneity: evolution through space and time. Cancer Res 72, 4875-4882, doi:10.1158/0008-5472.Can-12-2217 (2012).
17 Bashashati, A. et al. Distinct evolutionary trajectories of primary high-grade serous ovarian cancers revealed through spatial mutational profiling. J Pathol 231, 21-34, doi:10.1002/path.4230 (2013).
18 Castellarin, M. et al. Clonal evolution of high-grade serous ovarian carcinoma from primary to recurrent disease. J Pathol 229, 515-524, doi:10.1002/path.4105 (2013).
19 Schwarz, R. F. et al. Spatial and temporal heterogeneity in high-grade serous ovarian cancer: a phylogenetic analysis. PLoS Med 12, e1001789, doi:10.1371/journal.pmed.1001789 (2015).
20 Lambrechts, S. et al. Genetic heterogeneity after first-line chemotherapy in high-grade serous ovarian cancer. Eur J Cancer 53, 51-64, doi:10.1016/j.ejca.2015.11.001 (2016).
21 Latifi, A. et al. Cisplatin treatment of primary and metastatic epithelial ovarian carcinomas generates residual cells with mesenchymal stem cell-like profile. J Cell Biochem 112, 2850-2864, doi:10.1002/jcb.23199 (2011).
22 Thakur, B. & Ray, P. Cisplatin triggers cancer stem cell enrichment in platinum-resistant cells through NF-kappaB-TNFalpha-PIK3CA loop. J Exp Clin Cancer Res 36, 164, doi:10.1186/s13046-017-0636-8 (2017).
23 Wang, Y. et al. Epigenetic targeting of ovarian cancer stem cells. Cancer Res 74, 4922-4936, doi:10.1158/0008-5472.CAN-14-1022 (2014).
24 Steg, A. D. et al. Stem cell pathways contribute to clinical chemoresistance in ovarian cancer. Clin Cancer Res 18, 869-881, doi:10.1158/1078-0432.CCR-11-2188 (2012).
25 Yang, W. et al. Therapeutic Strategies for Targeting Ovarian Cancer Stem Cells. Int J Mol Sci 22, doi:10.3390/ijms22105059 (2021).
26 Zhang, S. et al. Identification and characterization of ovarian cancer-initiating cells from primary human tumors. Cancer Res 68, 4311-4320, doi:10.1158/0008-5472.CAN-08-0364 (2008).
27 Gao, M. Q., Choi, Y. P., Kang, S., Youn, J. H. & Cho, N. H. CD24+ cells from hierarchically organized ovarian cancer are enriched in cancer stem cells. Oncogene 29, 2672-2680, doi:10.1038/onc.2010.35 (2010).
28 Baba, T. et al. Epigenetic regulation of CD133 and tumorigenicity of CD133+ ovarian cancer cells. Oncogene 28, 209-218, doi:10.1038/onc.2008.374 (2009).
29 Silva, I. A. et al. Aldehyde dehydrogenase in combination with CD133 defines angiogenic ovarian cancer stem cells that portend poor patient survival. Cancer Res 71, 3991-4001, doi:10.1158/0008-5472.Can-10-3175 (2011).
30 Robinson, M. et al. Characterization of SOX2, OCT4 and NANOG in Ovarian Cancer Tumor-Initiating Cells. Cancers (Basel) 13, doi:10.3390/cancers13020262 (2021).
31 Li, Y. R. et al. Profiling ovarian cancer tumor and microenvironment during disease progression for cell-based immunotherapy design. iScience 26, 107952, doi:10.1016/j.isci.2023.107952 (2023).
32 Wang, L. et al. CAFs enhance paclitaxel resistance by inducing EMT through the IL‑6/JAK2/STAT3 pathway. Oncol Rep 39, 2081-2090, doi:10.3892/or.2018.6311 (2018).
33 Pasquier, J. et al. CCL2/CCL5 secreted by the stroma induce IL-6/PYK2 dependent chemoresistance in ovarian cancer. Mol Cancer 17, 47, doi:10.1186/s12943-018-0787-z (2018).
34 Wang, Y. et al. Autocrine production of interleukin-6 confers cisplatin and paclitaxel resistance in ovarian cancer cells. Cancer Lett 295, 110-123, doi:10.1016/j.canlet.2010.02.019 (2010).
35 Leung, C. S. et al. Cancer-associated fibroblasts regulate endothelial adhesion protein LPP to promote ovarian cancer chemoresistance. J Clin Invest 128, 589-606, doi:10.1172/jci95200 (2018).
36 Zhang, D. et al. Midkine derived from cancer-associated fibroblasts promotes cisplatin-resistance via up-regulation of the expression of lncRNA ANRIL in tumour cells. Sci Rep 7, 16231, doi:10.1038/s41598-017-13431-y (2017).
37 Gonzalez, V. D. et al. Commonly Occurring Cell Subsets in High-Grade Serous Ovarian Tumors Identified by Single-Cell Mass Cytometry. Cell Rep 22, 1875-1888, doi:10.1016/j.celrep.2018.01.053 (2018).
38 Xu, Y. et al. SUSD2 promotes cancer metastasis and confers cisplatin resistance in high grade serous ovarian cancer. Exp Cell Res 363, 160-170, doi:10.1016/j.yexcr.2017.12.029 (2018).
39 Feng, T., Wang, Y., Lang, Y. & Zhang, Y. KDM5A promotes proliferation and EMT in ovarian cancer and closely correlates with PTX resistance. Mol Med Rep 16, 3573-3580, doi:10.3892/mmr.2017.6960 (2017).
40 Sosulski, A. et al. CD44 Splice Variant v8-10 as a Marker of Serous Ovarian Cancer Prognosis. PLoS One 11, e0156595, doi:10.1371/journal.pone.0156595 (2016).
41 Alhabbab, R. Y. Targeting Cancer Stem Cells by Genetically Engineered Chimeric Antigen Receptor T Cells. Front Genet 11, 312, doi:10.3389/fgene.2020.00312 (2020).
42 Wolf, N. K., Kissiov, D. U. & Raulet, D. H. Roles of natural killer cells in immunity to cancer, and applications to immunotherapy. Nature Reviews Immunology 23, 90-105, doi:10.1038/s41577-022-00732-1 (2023).
43 Chovatiya, N. et al. Inability of ovarian cancers to upregulate their MHC-class I surface expression marks their aggressiveness and increased susceptibility to NK cell-mediated cytotoxicity. Cancer Immunol Immunother 71, 2929-2941, doi:10.1007/s00262-022-03192-7 (2022).
44 Kaur, K. et al. Super-charged NK cells inhibit growth and progression of stem-like/poorly differentiated oral tumors in vivo in humanized BLT mice; effect on tumor differentiation and response to chemotherapeutic drugs. OncoImmunology 7, e1426518, doi:10.1080/2162402X.2018.1426518 (2018).
45 Kaur, K. et al. Sequential therapy with supercharged NK cells with either chemotherapy drug cisplatin or anti-PD-1 antibody decreases the tumor size and significantly enhances the NK function in Hu-BLT mice. Frontiers in Immunology 14, doi:10.3389/fimmu.2023.1132807 (2023).
46 Kaur, K. et al. Probiotic-Treated Super-Charged NK Cells Efficiently Clear Poorly Differentiated Pancreatic Tumors in Hu-BLT Mice. Cancers 12, 63 (2020).